

# Inducing the Attachment of Cable Bacteria on Oxidizing Electrodes

Cheng Li[1], Clare E. Reimers[1], and Yvan Alleau[1]

[1]College of Earth, Ocean and Atmospheric Sciences, Oregon State University, Corvallis, Oregon 97331, USA

*Corresponding to*: Cheng Li (cheng.li@oregonstate.edu)

**Abstract.** The scope of the present study is to introduce electrochemical reactors as a tool for investigating the growth of novel filamentous cable bacteria and their unique extracellular electron transfer ability. New evidence that cable bacteria are widely distributed in sediments throughout an estuarine system connected to the NE Pacific Ocean is also presented. Cable bacteria found within Yaquina Bay, Oregon, USA, appear to cluster with the genus, *Candidatus*

*Electrothrix*. Results of a 135-day bioelectrochemical reactor experiment confirm a previous observation that cable bacteria can grow on oxidatively poised electrodes suspended in anaerobic seawater above reducing sediments. However, several diverse morphologies of *Desulfobulbaceae* filaments, cells, and colonies were observed on the carbon fibers of the suspended electrodes including encrusted chains of cells. These observations provide new information to suggest what conditions will induce cable bacteria to perform electron donation to an electrode surface,

further informing future experiments to culture cable bacteria apart from a sediment matrix.

## 1 Introduction

Long distance electron transfer (LDET) is a mechanism used by certain microorganisms to generate energy through the transfer of electrons over distances greater than a cell-length. These microorganisms may pass electrons across dissolved redox shuttles, nanofiber-like cell appendages, outer-membrane cytochromes, and/or mineral nanoparticles

to connect extracellular electron donors and acceptors (Li et al., 2017; Lovley, 2016). Recently, a novel type of LDET exhibited by filamentous bacteria in the family of *Desulfobulbaceae* was discovered in the uppermost centimeters of various aquatic, but mainly marine, sediments (Malkin et al., 2014; Pfeffer et al., 2012). These filamentous bacteria, also known as "cable bacteria", electrically connect two spatially separated redox half-reactions and generate electrical current over distances that can extend to centimeters, which is an order of magnitude longer than previously recognized

LDET distances (Meysman, 2017).

The unique ability of cable bacteria to perform LDET creates a spatial separation of oxygen reduction in oxic surface layers of sediment from sulfide oxidation in subsurface layers (Meysman, 2017). The spatial separation of these two half-reactions also creates localized porewater pH extremes in oxic and sulfidic layers, which induces a series of secondary reactions that stimulate the geochemical cycling of elements such as iron, manganese, calcium, phosphorus,

and nitrogen (Kessler et al., 2018; Rao et al., 2016; Seitaj et al., 2015; Sulu-Gambari et al., 2016b, 2016a). In addition to altering established perceptions of sedimentary biogeochemical cycling and microbial ecology (Meysman, 2017; Nielsen and Risgaard-Petersen, 2015), cable bacteria also possess intriguing structural features that may inspire new engineering applications in areas of bioenergy harvesting and biomaterial design (Lovley, 2016).



Much is still unknown about the basic ability of cable bacteria to perform LDET. It has been suggested that when long

filaments form, a chain of cells at the sulfidic terminal catalyzes anodic half reactions (½ H₂S + 2H₂O → ½ SO₄²⁻ + 4e⁻ + 5H⁺) while a cathodic half reaction (O₂ + 4e⁻ + 4H⁺ → 2H₂O) is catalyzed by cells at the oxic terminal. Electron transfer then occurs along the longitudinal ridges of cable bacteria filaments via electron hopping promoted by extracellular cytochromes positioned within a redox gradient (Bjerg et al., 2018; Meysman et al., 2015; Pfeffer et al., 2012). However, this hypothesis has not been directly verified and the electrical conductivity of cable bacteria

filaments or their longitudinal ridges has not been measured. Furthermore, cable bacteria remain uncultured and difficult to grow outside of sediment, complicating efforts to study them using different techniques, such as electrochemical assays and metatranscriptomics.

In a previous benthic microbial fuel cell (BMFC) experiment (Reimers et al., 2017), we serendipitously observed the attachment of cable bacteria to carbon fibers serving as an anode in an anaerobic environment. This finding suggested

that cable bacteria possess the ability to donate electrons to solid electron acceptors. While the understanding of cathodic potentials utilized by cable bacteria has been extended, further investigation is still needed to study the conditions that trigger the attachment of cable bacteria to a poised electrode in the laboratory and to provide information about the ability of cable bacteria to use an electrode as an electron acceptor. In the present study, we describe the design of a bioelectrochemical reactor configured to mimic the environment in the anodic chamber of a

BMFC and verify conditions that can induce cable bacteria attachment on electrodes. As part of this research, we also clarify the phylogenetic placement of cable bacteria found in sediments from Yaquina Bay, Oregon. Results assert that when oxygen is not available these cable bacteria can glide through different redox layers to an electrode poised at oxidative potentials. Thus, the present study provides new information about the chemotaxis of cable bacteria in environments other than sediments, revealing key conditions for their growth of cable bacteria in both natural and

engineered environments.

## 2 Materials and Methods

### 2.1 Study Site and Sediment Collection

Several studies suggest that cable bacteria may be found widely in coastal sediments possessing high rates of sulfide generation coupled with high rates of organic matter mineralization (Larsen and Nielsen, 2015; Malkin et al., 2014;

Pfeffer et al., 2012). Therefore to initiate this enquiry, sediment with these two characteristics was collected from Yaquina Bay, Oregon, USA using a hand shovel at a site on an intertidal mud flat (IMF, Latitude 44˚37'30 N, Longitude 124˚00'26 W), and secondly using a sediment grab to recover subtidal deposits from downstream of a commercial oyster farm (OFS, Latitude 44°34'37.0 N, Longitude 123°59'21.4 W) (Fig. 1). The IMF and OFS sites are located about 3 km and 7 km, respectively, upstream from the site where the BMFC was deployed in the

abovementioned study (Reimers et al., 2017). The top 20 cm of these sediments were sieved through a 0.5 mm mesh size metal screen to remove macrofauna and shell debris. Then the sieved sediments were allowed to settle and stored in sealed buckets in a cold room at 5˚C.



## 2.2 Sediment Incubation

To cultivate cable bacteria of Yaquina Bay, IMF and OFS sediments were each initially incubated for 60 days. These first incubations were started 2-5 days after collection and performed after homogenizing the sieved sediments under a flow of $N_2$ and then packing the sediment into triplicate polycarbonate tubes (15 cm height and 9.5 cm inner diameter). These cores were submerged in an aquarium containing aerated seawater collected from Yaquina Bay and held at 15°C, a temperature that is about average on the mudflats of Yaquina Bay (Johnson, 1980). Once a distinctive

suboxic layer was evident from color changes in the top centimeters of the cores, profiles of porewater pH, $O_2$, and $H_2S$ were measured to 2-3 cm depth with commercial microelectrodes (Unisense A.S., Aarhus, Denmark) to confirm geochemical evidence of cable bacteria activity (see below). Small subcores (0.5 cm diameter, 3 cm in length) were then taken out from each incubated core using cutoff syringes. The sediment plugs were washed gently to reduce the volume of fine particles, and cable bacteria biomass was further separated out from the sediment matrix by using

custom-made tiny glass hooks after Malkin et al. (2014). Washed sediment and separated biomass were frozen or fixed for subsequent phylogenetic and microscopic characterizations.

## 2.3 Reactor Configuration and Operation

To mimic the conditions where cable bacteria were found attached to electrode fibers in a BMFC (Reimers et al., 2017), a bioelectrochemical reactor was assembled from a polycarbonate core tube (15 cm height and 11.5 cm inner

diameter, Fig. 2) as a second phase of this research. A lid, a center rod to locate and support the electrodes, and a perforated bottom partition were made from polyvinyl chloride (PVC, McMaster-Carr, Elmhurst, IL). Three carbon brush electrodes that would serve as two anodes and a control electrode (Mill-Rose, Mentor, OH, 2 cm in diameter and 8.9 cm of total length) were inserted through septa within holes in the core lining to meet the center rod and were spaced radially at 120° angles from each other.

To initiate the experiment, the reactor was placed inside an 8L plastic beaker (with perforated walls) containing 3 cm of IMF sediments at the bottom. Enough additional IMF sediment was then placed inside the reactor to form an 8 cm thick layer after settling/compacting. In this configuration, the sediment-water interface was approximately 1 cm away from the lower extent of the carbon brush electrodes. The beaker was then gently lowered inside an aquarium filled with Yaquina Bay seawater until fully submerged, and the reactor was left uncapped. Seawater in the aquarium was

maintained at 15°C and bubbled to maintain air-saturation. A fuel cell circuit was completed by placing a 10 cm-long carbon-fiber brush cathode (Hasvold et al., 1997) and a reference electrode (Ag/AgCl [3 M KCl], MI-401F, Microelectrodes, Inc., Bedford, NH) into the seawater outside the reactor tube (Fig. 2).

The reactor was monitored in an open circuit state for 31 days to allow the development of a cable bacteria population within the top centimeters of sediment as had been observed in previous incubations. Microelectrode profiling was

used to characterize the vertical distribution of porewater pH and concentrations of $O_2$ and $H_2S$ on days 13 and 24 of incubation. On day 31, carbon fiber samples were trimmed off the unpoised anode brushes as control samples and the reactor was sealed to create fully anoxic conditions. Beginning on day 44, under seal, cathode versus anode potentials were poised at 300 mV by regulating two of the three anode carbon brushes with an individual custom-designed potentiostat circuit board (NW Metasystems, Bainbridge Island, WA) (Fig. 2). The third brush was kept at open circuit



as a continuing control. Electrode potentials of the anode (versus reference) and whole cell, and the current flow between anodes and cathode were monitored and recorded every 7 min with a multichannel datalogger (Agilent Technologies, Santa Clara, CA, model 34970A fitted with two 34901A multiplexer modules) wired to the potentiostat outputs. The electrodes were poised for more than 3 months. On day 135, they were extracted through the side openings in the bioreactor tube for SEM and CARD-FISH analyses (described below).

**2.4 Microelectrode Profiling**

The sediments incubated in open cores and in the bioelectrochemical reactor were each profiled with pH, $O_2$, and $H_2S$ microelectrodes to provide signature geochemical evidence of cable bacteria activity (Malkin et al., 2014). Microelectrodes had tip diameters of 100 μm. The $O_2$ microelectrodes were calibrated in air-purged seawater (as 100% air saturation) and in a solution of sodium ascorbate and NaOH (both to a final concentration of 0.1 M, as 0% $O_2$

saturation). Vertical oxygen microprofiles were recorded starting from 2 mm above either the sediment-water interface, or in the reactor above the carbon brush, at step size of 400 μm. Vertical pH and $H_2S$ microprofiles were measured concurrently at the same spatial interval. The pH microelectrode was calibrated by using standard pH 4, 7, and 10 buffer solutions (Ricca Chemical, Arlington, TX, United States). $H_2S$ microelectrodes were calibrated by generating an 11-point calibration relationship by standard addition, from 0 to 7.48 μM $H_2S$ at pH = 1.6. Total sulfide

concentration at each profile depth was derived from pH and $H_2S$ according to equilibrium relationships given in Millero et al. (1988) (Millero et al., 2018).

**2.5 SEM**

To confirm and examine the characteristic longitudinal ridges and cell-cell junctions of cable bacteria, filaments extracted from the sediments and carbon fibers from the reactor electrodes were visualized by scanning electron

microscopy (SEM). Samples were dehydrated in a graded series of ethanol solution from 10 to 100%. Specimens were then mounted on aluminum SEM stubs with double-sided carbon tape, critical-point dried using an EMS 850 Critical Point Dryer, and sputter-coated with gold and palladium using a Cressington 108 sputter coater. The resultant specimens were observed under a FEI Quanta 600FEG ESEM at 5–15 kV. This instrument also provided elemental spectra by X-Ray Energy Dispersive Spectrometry (EDS).

**2.6 CARD-FISH**

Catalyzed reporter deposition-fluorescence *in situ* hybridization (CARD-FISH) was used to microscopically identify *Desulfobulbaceae* filaments using a *Desulfobulbaceae*-specific oligonucleotide probe (DSB706; 5′-ACC CGT ATT CCT CCC GAT-3′) labelled with horseradish peroxidase (Lücker et al., 2007). In preparation for CARD-FISH, sediment samples were fixed with a 1:1 (vol: vol) ethanol and phosphate-buffered saline solution and stored at -20 °C

until analysis. Extracted bacterial filaments and carbon fibers cut from the carbon brush electrodes were treated with a fixative solution containing 1.25% glutaraldehyde and 1.3% osmium tetroxide. Fixed samples were stored at -20 °C until analysis. Sediment and bacterial filament samples were first retained on polycarbonate membrane filters and then mounted onto a glass slide by using 0.2 % agarose (Malkin et al., 2014). Carbon fiber samples were mounted directly





onto a glass slide without retaining on filter. Mounted samples were sequentially permeabilized by 10 mg/mL of lysosome (2 hrs at 37°C) and achromopeptidase (1 hr at 37°C). After permeablization, glass slides were incubated in $H_2O_2$ (0.15% in methanol) for 30 mins at room temperature (~25°C) to inactivate the endogenous peroxidases. The hybridization process was performed in a standard hybridization buffer at 46°C with 45% formamide for 7 hrs(Wendeberg, 2010). Alexa Fluro 488 (ThermoFisher, Waltham, MA, United States) was deposited on samples in the presence of 0.15% $H_2O_2$.

Two-color CARD-FISH was performed on some carbon fiber samples to look for a previously observed co-occurrence of cable bacteria and other electroactive bacteria on electrode surfaces (Reimers et al., 2017). To perform the two-color CARD-FISH, horseradish peroxidase on the hybridized DSB706 probes were inactivated by 0.15% $H_2O_2$. The inactivated samples were then hybridized with a *Desulfuromonadales*-specific oligonucleotide probe (DRM432; 5′-CTT CCC CTC TGA CAG AGC−3′) modified with horseradish peroxidase in standard hybridization buffer at 46°C

with 40% formamide for 5 hrs and sequentially stained with Alexa Fluro 555 (ThermoFisher, Waltham, MA, United States). A counter stain, 4′,6-diamidino-2-phenylindole (DAPI), was applied to all samples after the deposition of fluorescent probe(s). Hybridization samples were visualized using confocal laser scanning microscopy (CLSM) (LSM 780, Zeiss, Jena, Germany).

## 2.7 Microbial Community Characterizations

To investigate the phylogeny of the cable bacteria discovered in Yaquina Bay, genomic DNA was extracted from washed sediment samples and from tangles of bacterial filaments separated from sediments using a MoBio PowerSoil DNA Extraction Kit. To avoid insufficient cell lysis, all samples went through 5-7 freeze-thaw cycles before the use of the extraction kit (Roose-Amsaleg et al., 2001). Bacterial 16S rRNA genes were amplified by PCR with random primers 357wF (5′-CCTACGGGNGGCWGCAG-3′) and 785R (5′-GACTACHVGGGTATCTAATCC-3′).

Amplification and sequencing of DNA (Illumina MiSeq Reagent Kit v3, 2 × 300 bp) was performed by the Center of Genome Research and Biocomputing at Oregon State University. Sequences were processed using DADA2 (v.1.10) in R (3.5.0) as described in a previous study (Callahan et al., 2016). Sequences were aligned to the Silva SSU Ref NR database (v.132) and clustered into operational taxonomic units (OTUs) at 97% similarity. Representative sequences classified to the family of *Desulfobulbaceae* were tagged and aligned to 16S rRNA gene sequences from previously

identified cable bacteria (Trojan et al., 2016). A phylogenetic tree was constructed using RaxML with 1,000 bootstraps (Stamatakis, 2014). Sequences from this study were deposited to the Genbank's Sequence Read Archive (MK388690-MK388723).

## 3 Results and Discussion

### 3.1 Cable Bacteria Activity in the Sediments of Yaquina Bay

During the initial open incubations of IMF and OFS sediments, the top centimeter of each core changed from dark to light gray, and a layer of brownish mineral formed 2 mm the sediment-water interface to ~0.2 cm depth. These changes indicate the depletion of iron sulfides and the formation of iron oxy(hydroxide)s, respectively (Risgaard-



Petersen et al., 2015). Hallmarks of the activity of cable bacteria were documented by microelectrode profiling after 53 and 34 days of culture in IMF and OFS sediments, respectively. These hallmarks were a sulfide-free suboxic zone

and opposing pH extremes at approximately 0.2 cm and 1-1.5 cm deep (Fig. 3a & 3b). Although a faint smell of sulfide was detected during collection of the sediment, total sulfide concentrations detected by microelectrode profiling were low compared to previous studies of marine sediments hosting cable bacteria (roughly above 200 µmol/L) (Malkin et al., 2014). The pH minima within the sulfidic layers of sediment were also lower than observed previously (roughly around 6.5). It is likely that the relatively long times of our incubations contributed to these low values.

After the geochemical hallmarks of cable bacteria were observed, SEM performed on extracted sediment and bacterial filament samples revealed that cells within extracted filaments were 0.5 to 1.2 µm wide and 2 to 3 µm long (Fig. 4a, b, & c). Typical morphological features of cable bacteria including the longitudinal ridges and the cell-cell junctions were observed (Pfeffer et al., 2012). Certain filaments extracted from sediments were covered by heterogeneous coatings of mineral particles as was observed recently by Geerlings et al. (Fig 4c) (Geerlings et al., 2018). These

particles have similar elemental compositions to some authigenic clays (Sturz et al., 1998), showing enrichments of silicon, aluminum, magnesium, and iron. In our open incubation samples, some thinner filaments were also seen that displayed no obvious longitudinal ridges, although cell-cell junctions were still visible (Fig. 4b). Extracted filaments reacted positively to the DSB 706 probe and DAPI (Fig. 4d). Certain filaments seemed to be thicker (3-3.5 µm diameter) when visualized using FISH (SI Fig. 1a). Granules that appeared to be impenetrable to light were also

observed along filaments when viewed with a transmitted light microscope. Most of these granules had high affinity to DAPI stain (SI Fig. 1c). Similar granules reported in a recent study, were suggested to be intracellular polyphosphate granules (Geerlings et al., 2018). Together, these observations indicate that the population of the cable bacteria, widely distributed along the NE Pacific estuarine system of Yaquina Bay, Oregon, has characteristics very similar to those observed on other coasts.

**3.2 Phylogenetic Analysis of the Cable Bacteria in Yaquina Bay**

After identifying cable bacteria filaments through SEM and FISH, we also sequenced the 16S rRNA gene in genomic DNA extracted from separated aggregations of cable bacteria biomass from the IMF sediment, and from washed sediments from the top 3 cm of both IMF and OFS cores that were incubated (Fig. 4e). Sequencing of the 16s rRNA gene yielded 725 OTUs in DNA extracted from the aggregations of cable bacteria biomass, in which the family of

*Desulfobulbaceae* was predominately abundant (> 80%). Identified OTUs from the extracted biomass of cable bacteria from IMF sediments were predominately *Candidatus* Electrothrix. Sequencing of the 16s rRNA gene yielded 1520 and 1524 OTUs respectively in DNA extracted from the top 3 cm of the IMF and OFS sediments. Among these OTUs, 96 and 69 of them respectively from IMF and OFS can be assigned to the family of *Desulfobulbaceae*. When the most abundant OTU in the family of *Desulfobulbaceae* in all extracted DNA were aligned with a previously established

taxonomy framework of cable bacteria, they clustered with *Candidatus* Electrothrix, a candidate genus that has been recently proposed for the cable bacteria (Trojan et al., 2016). An OTU in OFS sediment clustered with the genus of *Candidatus* Electronema, which is associated with freshwater cable bacteria. As the OFS site is in the mid-estuary





zone of the Yaquina River, it is under the influence of low salinity seawater in winter, and therefore may be expected to have a community composition distinct from the IMF site.

Partial 16s rRNA sequences of cable bacteria have been discovered in sediment samples from the US east coast, Gulf of Mexico, and certain sites on the west coast from SILVA or GenBank databases (Trojan et al., 2016). Our studies have provided the first combined microscopic and genetic observations of cable bacteria in sediments from the NE Pacific coast of the United States, reinforcing the suggestion that these filamentous bacteria are distributed globally. This result also indicates that Yaquina Bay, OR, where we deployed the previous BMFC, indeed harbors a rich

population of cable bacteria.

### 3.3 Encouraging the Growth of Cable Bacteria on Poised Electrodes

We chose the IMF sediment to seed our bioelectrochemical reactor because these sediments developed a higher relative abundance of cable bacteria compared to the OFS sediment (data not shown) and their location was closer to the deployment site of the previous BMFC. Geochemical hallmarks of cable bacteria developed within two weeks of

culture within the bioelectrochemical reactor. The pH minimum at the sulfidic layer and pH maximum at the subsurface of sediment became more extreme at day 24, indicating that a cable bacteria population was actively mediating sulfide oxidation and transporting electrons to reduce oxygen. After sealing the reactor was closed, oxygen concentration in the overlaying seawater dropped below detection limits and the open circuit anode potential fell to more negative than -100 mV (vs Ag/AgCl). Once poised with the potentiostat, the cathode and anode potentials

became stable at ~330 mV and ~30 mV versus Ag/AgCl, respectively. Current collection started to increase once the anodic potential became stable, indicating that the anode brushes were being used as an electron acceptor. Current collected from duplicate electrodes was similar and each steadily increased to ~ 30.5 ± 2.5 µA by day 86, stabilized, then rose again to a peak of ~ 75 ± 8 µA on day 101. After this maximum, current decreased and restabilized at ~ 30 ± 5 µA. These electrochemical results are portrayed in Fig. 5. The cause of the current rise and subsequent fall (Fig.

5) is unknown but is a common occurrence in marine BMFC experiments (Nielsen et al., 2009; Ryckelynck et al., 2005). It is likely that such behavior is a result of a varying supply of reductants from the underlying sediment, changes in the anodic biofilm, and finally anode surface area loss caused by mineral deposition induced by microbial activity and the applied electrical potential. Coatings containing iron, phosphorus, sulfur, silicon, and aluminum are often found on anode surfaces of bioelectrochemical reactors in anoxic marine environments (Nielsen et al., 2008) and were

seen by SEM (see below). Cyclic voltammetry (CV, SI Fig. 2a) performed on the anode brushes at day 52 and 100 yielded broad and poorly defined electrochemical signals. The interpretation of such voltammograms may be complicated by a high uncompensated resistance between working electrode and reference electrode (Babauta and Beyenal, 2015a). While an oxidation peak can be clearly identified at potentials near where the anode was held, the peak current did not increase with an increase of scan rate (SI Fig. 2b). The peak oxidation current also did not change

much between day 52 and day 100. This behavior suggests that any current generated by the biomass of electroactive bacteria, including cable bacteria, was obscured during scans by current arising from irreversible redox reactions, such as oxidation of dissolved iron. The reduction peak was unidentifiable throughout the scan, a common phenomenon in benthic and sediment MFCs (Babauta and Beyenal, 2015b). Taken together, these results suggest that the electrode



### 3.4 Examining the Attachment of Cable Bacteria on the Anode

The hypothesis that led to the bioreactor experiments in this study was that an electrode poised at an oxidative potential can produce redox conditions and geochemical gradients that attract and signal the electron donation of cable bacteria. Several observations that were made on harvested electrodes affirm this hypothesis. First, under SEM, bacteria filaments with visible longitudinal ridges and cell-cell junctions were found integrated into biofilms on the surfaces of poised electrodes (Fig. 6a & b). Some filaments appeared to contain a smaller number of ridges (8 to 10) compared to previously reported cable bacteria filaments and others were without pronounced ridges along their longitudinal axes. The latter examples did show cell-cell junctions and appeared to have wrinkled surfaces that were similar to the thin cable bacteria filaments extracted from our cultured sediment cores (Fig. 6c & 6d). Secondly, many bacteria filaments on electrode surfaces were encrusted suggesting mineral deposition similar to that observed at the oxic terminal of cable bacteria in sediments (Fig. 6e & 6f). EDS indicated that these deposits contained iron, phosphorus, oxygen, and silicon (SI Fig. 3b). The control electrode that was not positively poised displayed no mineral deposition and nearly no cell growth (Fig. 6). Third, many of the bacterial filaments on the poised electrode surfaces reacted positively with the *Desulfobulbaceae*-specific probe. CARD-FISH performed in the present study suggested that the anodic carbon fibers harbored many short bacterial filaments and colonies belonging to the family of *Desulfobulbaceae* (Fig. 7a, b, c, d, e, f). Clear cell-cell junctions were observed along many of the fluorescent filaments. However, the complexity of the carbon fiber samples often hampered clear microscopic visualization. Application of an additional *Desulfuromonadales*-specific oligonucleotide probe (DRM432) confirmed the presence of other known electrogenic bacteria on the carbon fibers near *Desulfobulbaceae* cells as well (Fig. 7 b, c & d).

Though a global occurrence has recently been indicated (Malkin et al., 2014), cable bacteria successfully evaded microbiological survey for quite a long time. One of the reasons is likely that the phylogeny of cable bacteria is shadowed under the family of *Desulfobulbaceae*, which are often highly abundant in marine sediments (Kuever, 2014). Another reason may be a resistance of the cells of cable bacteria to routine cell lysing techniques that have been used with many DNA extraction kits (Trojan et al., 2016). Therefore, the observations of the unique filamentous form and morphological features (ridges and cell-cell junctions) of cable bacteria, combined with fluorescence *in situ* hybridization labeling have been employed to identify their presence in various studies (Malkin et al., 2014, 2017; Malkin and Meysman, 2015). The electrochemical reactor was anoxic for more than 100 days. The observation of cable bacteria on the anodic carbon fibers confirmed that they can survive under such conditions and were likely using the anode as electron acceptor as suggested previously (Reimers et al, 2017). The closest culturable relative, *Desulfobulbus propionicus*, can utilize an electrode as an electron acceptor to oxidize $S^0$, $H_2$, and organic acids like pyruvate, lactate, and propionate (Holmes et al., 2004). While cable bacteria appear to possess features like motility and an ability to form loops and bundles that are similar to large sulfur bacteria (but distinct from the *D. propionicus*), our SEM and CARD-FISH examinations suggest that cable bacteria on surfaces may produce extracellular structures to transfer electrons to an electrode and/or to insoluble Fe(III)-oxides similar to *D. propionicus* (Bjerg et al., 2016;





Holmes et al., 2004; Jørgensen, 2010; Pfeffer et al., 2012). Admittedly, indisputable proof of electron transfer from cable bacteria to electrodes still awaits growth in purer biofilms and cultures.

The present study demonstrated the possibility of drawing cable bacteria out of sediments for further culture on an electrode. Bjerg et al. (2016) have suggested cable bacteria form twisted loops to move out of sediment when oxygen becomes unavailable. How the cable bacteria adjust themselves when the loop lands on a new location where electron

acceptors and donors are available remains unknown. In the microenvironment induced by the poised anode, assuming the cable bacteria can use electrode as an electron acceptor, the distance between the electron donor and acceptor utilized by cable bacteria may be short, reducing the advantage of long filaments. Recently, Aller et al. (2019) discovered the presence of short cable bacteria filaments in a bioturbated zone associated with the tube worm *Chaetopterus variopedatus* likely as a response to the redox microenvironment, prompting a similar suggestion about

what may control filament length and morphologies. Additionally, a change of electron transfer mechanism could alter the gene expression, as have been suggested in *Geobacter sulfurreducens* growing at different electrical potentials (Malvankar et al., 2011; Yi et al., 2009). In summary, when growing on an electrode poised at an oxidative potential, cable bacteria may no longer require long filaments or be able to maintain them due to the nature of the potential gradient.

**4 Summary and Implications for the Electrode Associated Growth of Cable Bacteria**

The present study introduces electrochemical reactors as a unique approach for investigation of filamentous cable bacteria and their unique ability to transfer electrons. In addition, we confirm that an active population of the filamentous cable bacteria are widely distributed along the estuary of Yaquina Bay, Oregon. Cable bacteria OTUs found in Yaquina Bay clustered closely with the genus of *Candidatus* Electrothrix, contributing a new location to the

global distribution of cable bacteria. Moreover, by incubating intertidal sediment collected from Yaquina Bay in a bioreactor mimicking the anodic chamber of a BMFC, we observed that this group of bacteria can be drawn to electrodes at oxidative electrical potentials and that they likely will use an electrode as an electron acceptor in the absence of dissolved oxygen (Nielsen et al., 2008; Reimers et al., 2017). Beside identified cable bacteria, filaments and cells within the family of *Desulfobulbaceae* that possessed different morphologies were also observed on the

anode surface, suggesting that cable bacteria may be able to alter their morphologies depending on the redox microenvironment. Heavy mineral encrustation observed on filaments attached to electrode was similar to that reported as occurring at the oxic terminal of cable bacteria in other marine sediments, suggesting that an electrode poised at an oxidative potential creates a similar redox environment to that created by oxygen diffusion and reaction at the interface of sulfidic sediments. However, more work is needed to determine potentials and conditions that will

not also lead to mineral precipitation on electrode surfaces. Cable bacteria greatly influence sediment habitats by performing electrogenic sulfur oxidation. Developing *ex situ* culture techniques and gaining insight into their electron transfer will contribute to the overall understanding of this group of bacteria and their survival in both natural and engineered environments.



**Author contribution**

CL and CR conceived the presented idea. YA designed and assembled the bioelectrochemical reactor. CL performed the microscopic examination, analysed the microbial community and phylogenies, and wrote the manuscript.

**Acknowledgment**

This research was funded through grant N00014-17-1-2599 from the Office of Naval Research to CR. We thank Teresa Sawyer, Electron Microscopy Facility Instrument Manager, Oregon State University for assistance with the

SEM imaging, and Anne-Marie Girard for valuable advice on the confocal microscope imaging. The authors wish to acknowledge the Confocal Microscopy Facility of the Center for Genome Research and Biocomputing at Oregon State University which is supported in part by award number 1337774 from the National Science Foundation. We also thank our lab members for providing critical comments on the manuscript.

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

**Figure 1. Detailed satellite image of Yaquina Bay estuary with study locations superimposed. Insert: North America**
**topographic map indicating the location of Newport, Oregon. IMF: intertidal mud flat sediment, OFS: oyster farm sediment,**
       **and BMFC: previous benthic microbial fuel cell deployment site.**

       **Figure 2. Schematic of the bioelectrochemical reactor design used in this study: (a) lateral view of the reactor; (b) birds eye**
       **view of the reactor cap and electrode arrangement; and (c) electrical circuit of the reactor. Dimensions are in cm. $A_1$ and**
       **$A_2$ represent the current monitored in duplicate anodes; $V_1$ represents the potential monitored between the duplicate anodes**
**and cathode; and $V_2$ and $V_3$ represent potentials monitored between the duplicate anodes and the reference electrode.**

       **Figure 3. Representative microelectrode depth profiles of oxygen (blue), pH (red), and $\Sigma H_2S$ (yellow) in (a) IMF and (b)**
       **OFS sediments after 53 and 34 days of incubation, respectively; and in the bioelectrochemical reactor at (c) day 13, and (d)**
       **day 24.**

       **Figure 4. Cable bacteria filaments in Yaquina Bay sediments. (a) A cable bacteria filament under SEM. (b) A thin type of**
**cable bacteria filament under SEM. (c) Multiple filaments of cable bacteria under SEM. Blue arrow indicates a thick type**
       **of cable bacteria and red arrow indicates a cable bacteria filament incorporated in the observed mineral coating. (d)**
       **Identification of the filaments belonging to *Desulfobulbaceae* using Catalyzed reporter deposition-fluorescence *in***
       ***situ* hybridization (DSB 706 probe + Alexa Fluor 488, green and DSB DAPI, blue). (e) Phylogenetic tree of *Desulfobulbaceae***
       **16s rRNA gene sequences recovered from IMF and OFS sediment samples and extracted biomass from IMF. Color boxes**
**indicate previously recognized species of cable bacteria. Scale bar shows 5% sequence divergence.**

       **Figure 5. The current production (blue), the anodic potential (black), and cathodic potential (orange) over time during the**
       **reactor experiment. The reference electrode was an Ag/AgCl electrode with saturated KCl filling solution. Figure only**
       **shows one of the duplicate electrodes.**

       **Figure 6. SEM images illustrating (a) & (b) cable bacteria filaments with visible ridges and cell-cell junctions incorporated**
**into the biofilms on carbon fiber electrode surfaces. Red triangles indicate the cell-cell junctions. (c) & (d) Long bacterial**
       **filaments without typical morphological features of cable bacteria. (e) & (f) Mineral encrusted bacterial filaments. (g) & (h)**
       **Colonies of long cells. (i) Image of control electrode surface after culture. Yellow arrows indicate the locations of long cells.**

       **Figure 7. (a), (b), and (c) Confocal microscope images illustrating cable bacteria filaments on the carbon fibers that served**
       **as an anode. (d), (e), & (f) Colonies of cells belonging to *Desulfobulbaceae*. Red circles indicate possible doublet of the long**
**cells. Cells were visualized using Catalyzed reporter deposition-fluorescence *in situ* hybridization (DSB 706 probe + Alexa**
       **Fluor 488, green; DRM 432 + Alexa Fluor 555, red; and DAPI, blue).**



**Fig. 1**







**Fig. 2**

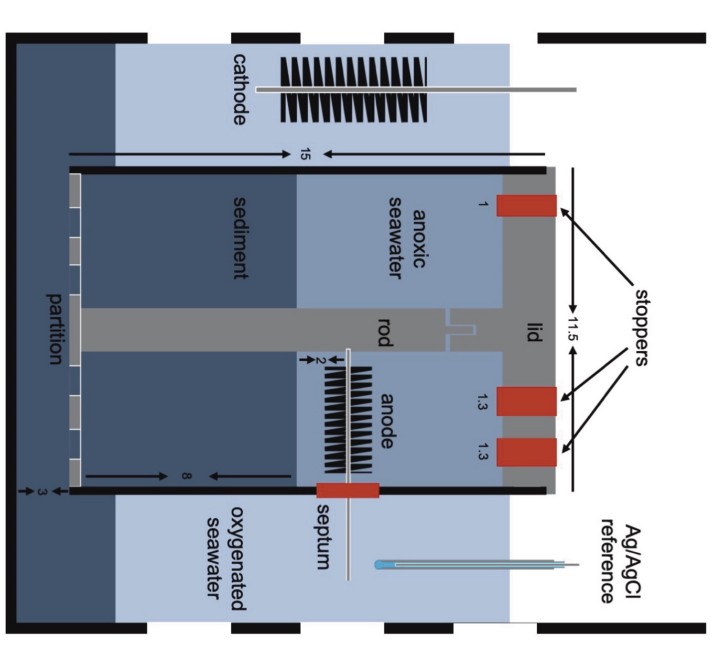

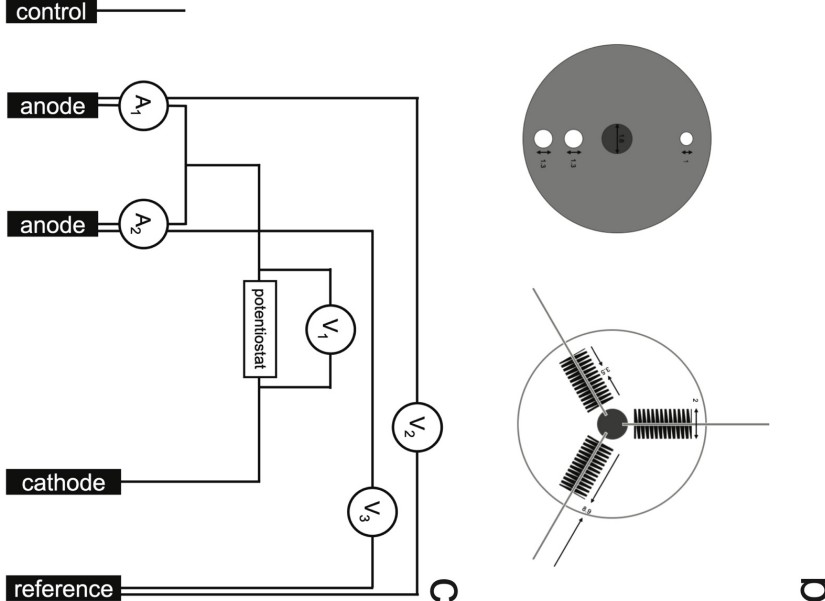



**Fig. 3**

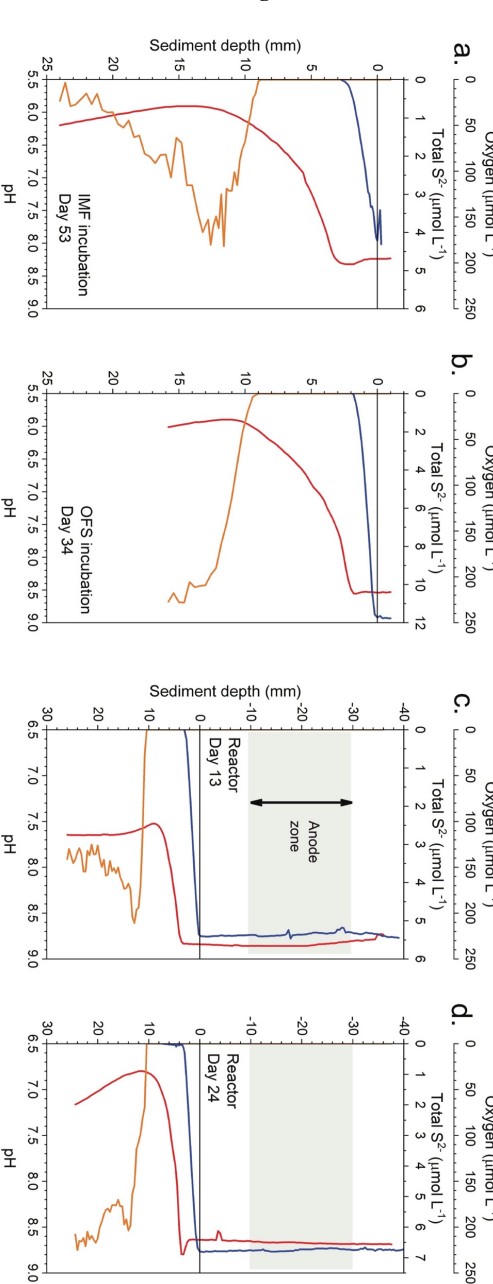




**Fig. 4**





**Fig. 5**

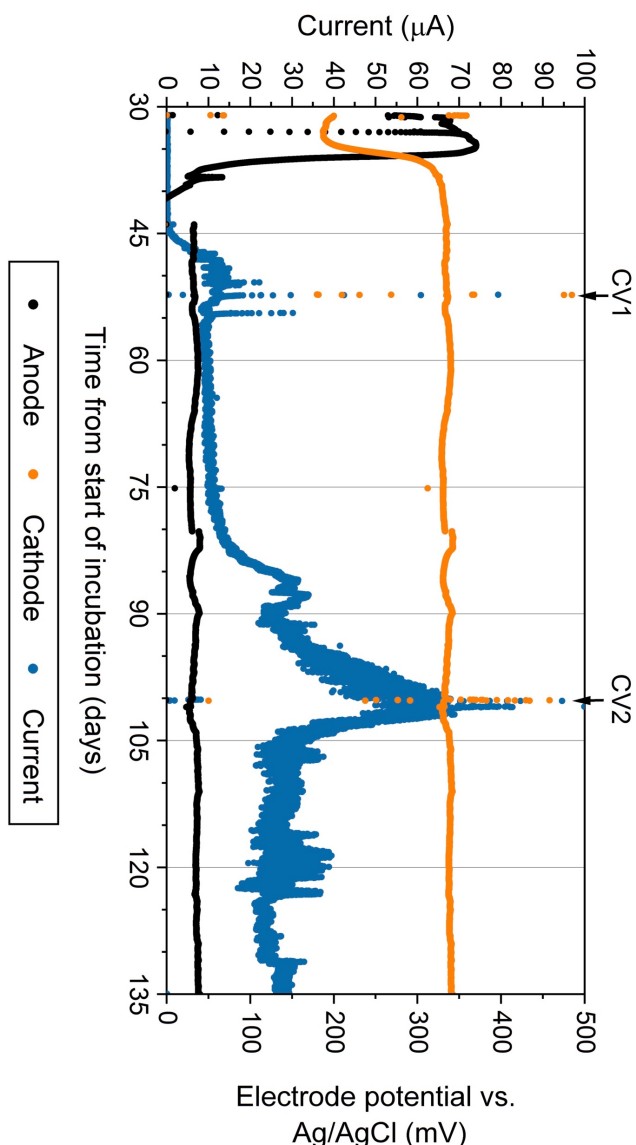




**Fig. 6**

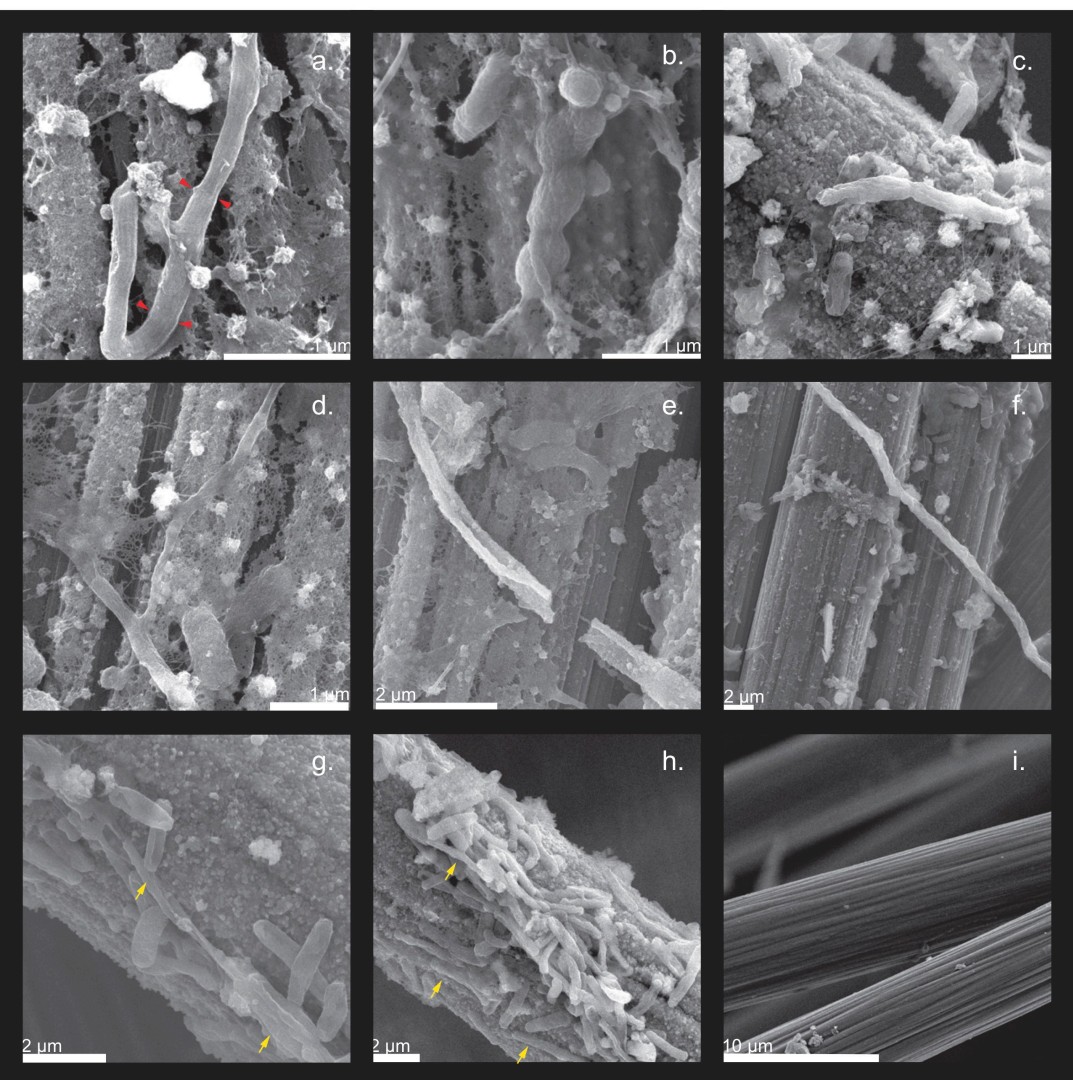





**Fig. 7**

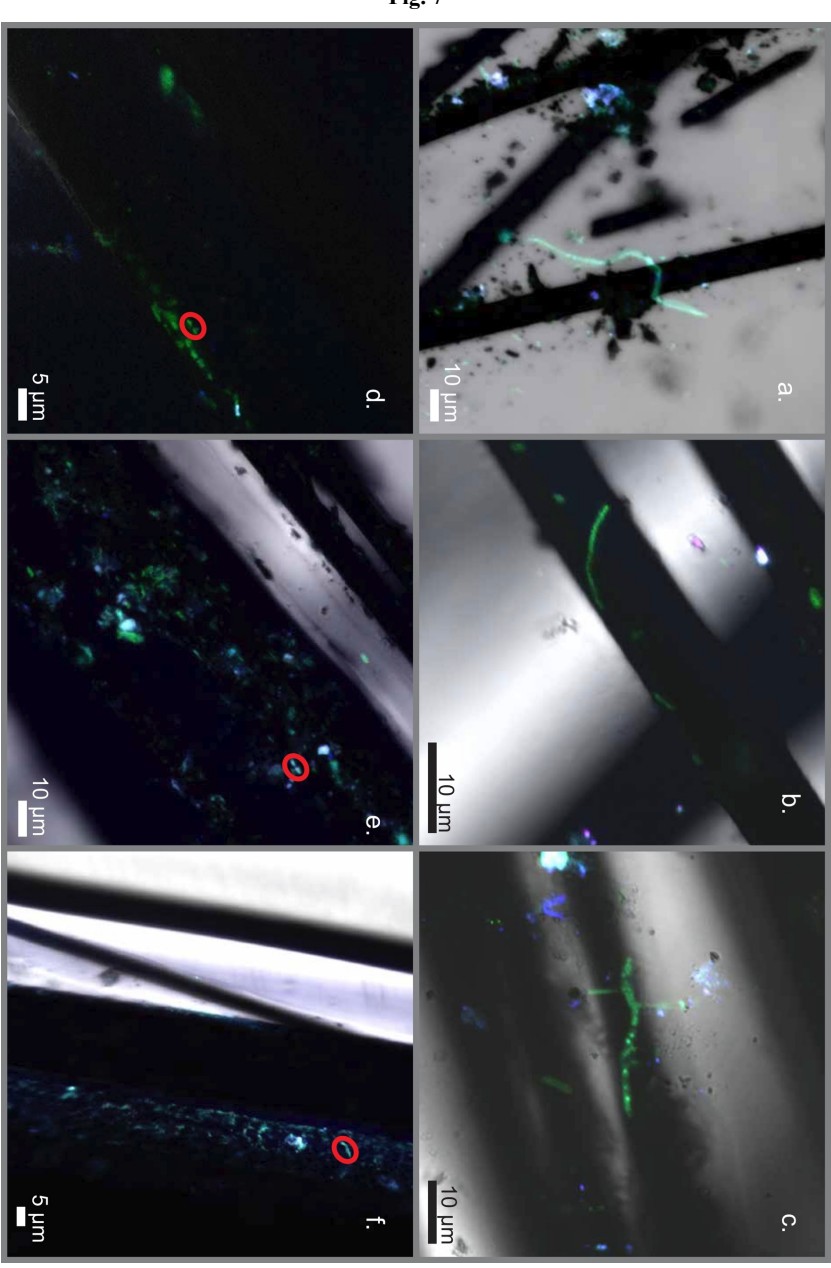