# Peer review of "Inducing the Attachment of Cable Bacteria on Oxidizing Electrodes"

_Biogeosciences, 2019_

## Referee Comment (RC1) · Anonymous Referee #1 · 16 Sep 2019

The manuscript is well written and gives clear messages. The research is well done. However, I am wondering what controls are used? Such as is there any electrode without polarization to see if the cable bacteria still grow. I believe discussion of control would critically improve the manuscript. I am wondering, what would happen the shape of the profiles (DO, S2-, and pH) if the polarization was stopped. Even without these controls, the manuscript is critically important to advance our knowledge on cable bacteria. I believe, this manuscript will generate many new research questions.

---

## Author Comment (AC1) · 24 Sep 2019

Thanks for your comments. All evidence supports that the cable bacteria grew and attached on the electrode may perform electron transfer with the electrode, though no direct observation was made. The complicated environment on the electrode surface (e.g. the deposited minerals and other types of electrode-associated bacteria) had hindered such observation. We will try to observe the growth and electron exchange of cable bacteria with the electrode in a more controlled environment in our coming experiment.

---

## Referee Comment (RC2) · Anonymous Referee #2 · 29 Sep 2019

The study of Li et al. is a follow up study on the Reimers et al. (20xx) where cable bacteria attachments to the anodes in microbial fuel cells were reported. Li and co-workers aim to reproduce the observations by establishing a microbial fuel cell in the lab and then investigate if cable bacteria attach to the anode . After 135 days of incubation cable bacteria attachments were observed through SEM imaging and CARD FISH. In addition to the primary work Li and Co-workers report the presence of cable bacteria at their study site Yaquina Bay by means of pH, O2 and H2S profiling, SEM, FISH and 16sRNA analysis. In general I think that the overall aim of this study is only loosely approached. The story goes in many directions and is not well focused: There are two lines one is the MCF line another is the report on cable bacteria in Yaquina Bay. If the primary aim was to investigate if cable bacteria can grow on anodes, why not

tone the latter story line a bit more down to avoid confusions about the experimental goals? It is drawback that the authors do not present quantitative estimates of cable bacteria density on the electrodes and that they only present SEM images. I thing that it would be better and more convincing with FISH or molecular tools (qpcr) that allows for both identification and quantification of cable bacteria on the anodes and on the control electrodes. This could allowed for a more robust comparison of the two types of systems and thus stronger conclusions. The techniques were used in the sediment studies, why where they not applied on in the experiment? Some of the citations are incorrect e.g. Risgaard-Petersen et al. 2015 is cited for observations that cable bacteria can deplete iron sulfide, but this paper report the discovery of cable bacteria in freshwater sediment. Should be Risgaard-Petersen 2012. Bjerg et al. 2016 and Pfeffer et al. 2012 are cited to document that D. propionicus can transfer electrons to electrode and/or to insoluble Fe(III)-oxides.This was not shown in these papers, which are on cable bacteria motility and on the discovery of the cable bacteria. Some statements are highly speculative and not supported by the presented data (l 293) "In summary, when growing on an electrode poised at an oxidative potential, cable bacteria may no longer require long filaments or be able to maintain them due to the nature of the potential gradient" There are no data in the study that can document such statement.

---

## Author Comment (AC2) · 19 Oct 2019

We sincerely apologize that we did not how to reply comment formly. Now we submit our final response as instructed by the editor.

Comment from Reviewer #1: Can cable bacteria exchange electrons with electrodes? The manuscript is well written and gives clear messages. The research is well done. However, I am wondering what controls are used? Such as is there any electrode without polarization to see if the cable bacteria still grow. I believe discussion of control would critically improve the manuscript. I am wondering, what would happen the shape of the profiles (DO, S2-, and pH) if the polarization was stopped. Even without these controls, the manuscript is critically important to advance our knowledge of cable

bacteria. I believe, this manuscript will generate many new research questions.

Response: We thank the reviewer for such positive comments. Our experimental results showed that cable bacteria migrated out of sediments and attached to poised electrodes where they most likely contributed to measured currents, implying they can perform electron transfer to an electrode. However, since they were only part of a mixed-species biofilm we cannot say definitively that this experiment shows they can exchange electrons with electrodes. The complicated environment on the electrode surface (e.g. the deposited minerals and other types of electrode-associated bacteria) hindered such a certain conclusion.

In our experimental setup, one of the 3 electrodes inside of the anodic chamber was maintained at the open circuit as a control electrode (as is stated in the section on Reactor Configuration and Operation). Scanning electron microscopy showed that the surface of this control electrode surface stayed relatively clean without any filamentous bacteria biomass or mineral deposition (Fig. 6i). We did conduct profiling with microelectrodes in the reactor after the anodes were poised but, in the process, broke our pH microelectrode. The profiles of $O_2$ and $H_2S$ were predictable due to the imposed anoxia and the presence of the anodes as a high-area oxidizing surface: dissolved oxygen was below detection in the overlying seawater and sulfide concentrations were also below the detection limit in overlying seawater but could be detected when the microelectrode entered sediment surface. We decided not to include this profile data because it was not very visual (zeros) and does not reveal what was happening on the electrode surface. When we dissembled the reactor, the seawater inside of the reactor had a pH near 6.2.

Changes in manuscript: This comment has given us a clear direction for the revision of our manuscript. We will revise the implication and conclusion sections to clearly state how the results from this study and other published research only indicate cable bacteria may exchange electrons with electrodes, and we will point out potential directions for upcoming experiments to observe definitively cable bacteria's electron transfer to

electrodes. We will also revise the Results and Discussion to give more information about microprofiling results in the closed reactors.

---

## Author Comment (AC3) · 19 Oct 2019

Comment from Reviewer #2: The study of Li et al. is a follow up study on the Reimers et al. (20xx) where cable bacteria attachments to the anodes in microbial fuel cells were reported. Li and coworkers aim to reproduce the observations by establishing a microbial fuel cell in the lab and then investigate if cable bacteria attach to the anode. After 135 days of incubation cable bacteria attachments were observed through SEM imaging and CARD FISH. In addition to the primary work Li and Co-workers report the presence of cable bacteria at their study site Yaquina Bay by means of pH, O2 and H2S profiling, SEM, FISH and 16sRNA analysis. In general, I think that the overall aim of this study is only loosely approached. The story goes in many directions and is not well focused: There are two lines one is the MCF line another is the report on cable

bacteria in Yaquina Bay. If the primary aim was to investigate if cable bacteria can grow on anodes, why not tone the latter story line a bit more down to avoid confusions about the experimental goals? It is drawback that the authors do not present quantitative estimates of cable bacteria density on the electrodes and that they only present SEM images. I think that it would be better and more convincing with FISH or molecular tools (qpcr) that allows for both identification and quantification of cable bacteria on the anodes and on the control electrodes. This could allow for a more robust comparison of the two types of systems and thus stronger conclusions. The techniques were used in the sediment studies, why where they not applied on in the experiment? Some of the citations are incorrect e.g. Risgaard-Petersen et al. 2015 is cited for observations that cable bacteria can deplete iron sulfide, but this paper report the discovery of cable bacteria in freshwater sediment. Should be Risgaard-Petersen 2012. Bjerg et al. 2016 and Pfeffer et al. 2012 are cited to document that D. propionicus can transfer electrons to electrode and/or to insoluble Fe (III)-oxides. This was not shown in these papers, which are on cable bacteria motility and on the discovery of the cable bacteria. Some statements are highly speculative and not supported by the presented data (l 293) "In summary, when growing on an electrode poised at an oxidative potential, cable bacteria may no longer require long filaments or be able to maintain them due to the nature of the potential gradient" There are no data in the study that can document such statement.

Response: We appreciate the reviewer's critical and insightful comments. Firstly, we agree with the reviewer that the aim of study needs to be made clearer by focusing on the reactor experiment rather than the broader characterizations of incubations from Yaquina Bay, Oregon. This part of the study was conducted to confirm that the sediment within Yaquina Bay can harbor a population of cable bacteria, and it will be toned down in revision.

Secondly, we also agree with the reviewer that a quantitative estimation of cable bacteria density coupled with SEM images would ideally be useful for a convincing argument

that cable bacteria can be attracted to an oxidizing electrode. However, in the present study, the density was quite low and thus CARD-FISH along with morphological observation by SEM were necessary to confirm the presence of cable bacteria on the electrodes. The SEM method was considered key, since no other member in the family of Desulfobulbaceae forms filaments with ridges along their longitudinal axis. More important than showing a high density, at the present stage of the study, we were trying to reconfirm that the cable bacteria can be drawn from sediments to grow on an oxidizing electrode and to deliver critical information about the conditions that trigger the attachment. Quantitative analyses such as qPCR will be employed in our next stage experiment that utilizes an electrochemical reactor without sediment to quantify the cable bacteria abundance on an electrode.

Thirdly, we thank the reviewer for pointing out our mistakes when sorting references. We did mean to cite Risgaard-Petersen 2012, Bjerg et al. 2016 and Pfeffer et al. 2012 in our manuscript. We cited Holmes et al. 2004 to indicate that D. propionicus can utilize an electrode as an electron acceptor to oxidize S0, H2, and organic acids like pyruvate, lactate, and propionate.

Lastly, the statement pointed out by the reviewer is speculative, though this statement was deduced from observations from our study and another study by Aller et al. 2019. We will make this statement less speculative in revision by only commenting on the observations of Aller et al.

Changes in manuscript: Firstly, we will tone down the presentation of the sediment core incubations from Yaquina Bay as suggested by the reviewer. In the revised manuscript, we will only present incubation and phylogenetic results from samples from the intertidal mudflat sediment which was used in the reactor incubation (eliminating the OFS site from figures and discussion). We will maintain the focus on determining if cable bacteria can grow on anodes and conditions in the anodic chamber that may trigger electrode attachment. Secondly, additional quantitative analyses will not be performed as an addition to the present manuscript. Such analyses would not be fruitful since

the cable bacteria density was low and affected by the mineral precipitation observed on the poised electrodes. Thirdly, we will carefully reexamine our reference to make sure that each citation accurately represents preceding statements. We will also add new references and revise the introduction to keep our information about the current cable bacteria research up-to-date. Lastly, we will revise the speculative statement pointed out by the reviewer. We will integrate observations from our study with those presented by Aller et al. 2019 without further speculation.

Please also note the supplement to this comment:
https://www.biogeosciences-discuss.net/bg-2019-334/bg-2019-334-AC3-supplement.pdf

---

## Author Response (AR1)

**College of Earth, Ocean, and Atmospheric Sciences**
Oregon State University
CEOAS Admin Bldg.
Corvallis, Oregon 97331-5503

**P** 541-737-3504
**F** 541-737-2064
ceoas.oregonstate.edu

11/17/19

Resubmission of the manuscript "Inducing the Attachment of Cable Bacteria on Oxidizing Electrodes (bg-2019-334)"

Dear Editor,

Thank you for the opportunity to let us revise our manuscript: "**Inducing the Attachment of Cable Bacteria on Oxidizing Electrodes (bg-2019-334)**". We appreciate the reviewers' comments, encouragements, and suggestions to our manuscript. We believe that the revised manuscript is substantially improved after making edits according to reviewers' comments.

Following this letter are our point-to-point response to reviewers' comments including how and where our manuscript has been modified and a marked-up manuscript version showing the changes made in our manuscript.

Thank you for consideration of this manuscript for publication.

Sincerely,

Cheng Li, Ph.D.

| Comments from reviewer #1 | Authors response | Changes in manuscript |
|---|---|---|
| Can the cable bacteria exchange electrons with electrodes? The manuscript is well written and gives clear messages. The research is well done. | We thank the reviewer for such positive comments. Our experimental results showed that cable bacteria migrated out of sediments and attached to poised electrodes where they most likely contributed to measured currents, implying they can perform electron transfer to an electrode. However, since they were only part of a mixed-species biofilm we cannot say definitively that this experiment shows they can exchange electrons with electrodes. The complicated environment on the electrode surface (e.g. the deposited minerals and other types of electrode-associated bacteria) hindered such a certain conclusion. | This comment has given us a clear direction for the revision of our manuscript. We will revise the implication and conclusion sections to clearly state how the results from this study and other published research only indicate cable bacteria may exchange electrons with electrodes, and we will point out potential directions for upcoming experiments to observe definitively cable bacteria's electron transfer to electrodes.

Line 359 to 390 |
| However, I am wondering what controls are used? Such as is there any electrode without polarization to see if the cable bacteria still grow. I believe discussion of control would critically improve the manuscript. I am wondering, what would happen the shape of the profiles (DO, S2-, and pH) if the polarization was stopped. Even without these controls, the manuscript is critically important to advance our knowledge on cable bacteria. I believe, this manuscript will generate many new research questions. | In our experimental setup, one of the 3 electrodes inside of the anodic chamber was maintained at open circuit as a control electrode (as is stated in the section on Reactor Configuration and Operation). Scanning electron microscopy showed that the surface of this control electrode surface stayed relatively clean without any filamentous bacteria biomass or mineral deposition (Fig. 5i). We did conduct profiling with microelectrodes in the reactor after the anodes were poised but, in the process, broke our pH microelectrode. The profiles of $O_2$ and $H_2S$ were predictable due to the imposed anoxia and the presence of the anodes as a high-area oxidizing surface: dissolved oxygen was below detection in the overlying seawater and sulfide concentrations were also below detection limit in overlying seawater but could be detected when the microelectrode entered sediment surface. We decided not to include this profile data because it was not very visual (zeros) and does not reveal what was happening on the electrode surface. When we dissembled the | We will also revise the Results and Discussion to give more information about microprofiling results in the closed reactors.

Line 139 to 144
Line 272 to 276
Also see Fig. 2d |

| | reactor, seawater inside of the reactor had a pH near 6.2. | |
| --- | --- | --- |
| **Comments from reviewer #2** | **Authors response** | **Changes in manuscript** |
| The study of Li et al. is a follow up study on the Reimers et al. (20xx) where cable bacteria attachments to the anodes in microbial fuel cells were reported. Li and coworkers aim to reproduce the observations by establishing a microbial fuel cell in the lab and then investigate if cable bacteria attach to the anode. After 135 days of incubation cable bacteria attachments were observed through SEM imaging and CARD FISH. In addition to the primary work Li and Co-workers report the presence of cable bacteria at their study site Yaquina Bay by means of pH, O2 and H2S profiling, SEM, FISH and 16sRNA analysis. In general, I think that the overall aim of this study is only loosely approached. The story goes in many directions and is not well focused: There are two lines one is the MCF line another is the report on cable bacteria in Yaquina Bay. If the primary aim was to investigate if cable bacteria can grow on anodes, why not tone the latter story line a bit more down to avoid confusions about the experimental goals? | We appreciate the reviewer's critical and insightful comments. Firstly, we agree with the reviewer that the aim of study needs to be made clearer by focusing on the reactor experiment rather that the broader characterizations of incubations from Yaquina Bay, Oregon. This part of the study was conducted to confirm that the sediment within Yaquina Bay can harbor a population of cable bacteria, and it will be toned down in revision. | Firstly, we will tone down presentation of the sediment core incubations from Yaquina Bay as suggested by the reviewer. In the revised manuscript, we will only present incubation and phylogenetic results from samples from the intertidal mudflat sediment which was used in the reactor incubation (eliminating the OFS site from figures and discussion). We will maintain the focus on determining if cable bacteria can grow on anodes and conditions in the anodic chamber that may trigger electrode attachment.

Line 6 to 23
Line 72 to 77
Line 91 to 93
Line 204 to 260 (section 3.1)
Line 359 to 390

Also see Fig. 2 and Fig. 3 |
| It is drawback that the authors do not present quantitative estimates of cable bacteria density on the electrodes and that they only present SEM images. I think that it would be better and more convincing with FISH or molecular tools (qpcr) that allows for both identification and quantification of cable bacteria on the anodes and on the control electrodes. This could allow for a more robust comparison of the two types of systems and thus stronger | Secondly, we also agree with the reviewer that a quantitative estimation of cable bacteria density coupled with SEM images would ideally be useful for a convincing argument that cable bacteria can be attracted to an oxidizing electrode. However, in the present study the density was quite low and thus CARD-FISH along with morphological observation by SEM were necessary to confirm the presence of cable bacteria on the electrodes. The SEM method was considered | Secondly, additional quantitative analyses will not be performed as an addition to the present manuscript. Such analyses would not be fruitful since the cable bacteria density was low and affected by the mineral precipitation observed on the poised electrodes.

No change was proposed |

| | | |
|---|---|---|
| conclusions. The techniques were used in the sediment studies, why where they not applied on in the experiment? | key, since no other member in the family of *Desulfobulbaceae* forms filaments with ridges along their longitudinal axis. More important than showing a high density, at present stage of the study, we were trying to reconfirm that the cable bacteria can be drawn from sediments to grow on an oxidizing electrode and to deliver critical information about the conditions that trigger the attachment. Quantitative analyses such as qPCR will be employed in our next stage of experiment that utilizes electrochemical reactor without sediment to quantify the cable bacteria abundance on electrode. | |
| Some of the citations are incorrect e.g. Risgaard-Petersen et al. 2015 is cited for observations that cable bacteria can deplete iron sulfide, but this paper report the discovery of cable bacteria in freshwater sediment. Should be Risgaard-Petersen 2012. Bjerg et al. 2016 and Pfeffer et al. 2012 are cited to document that D. propionicus can transfer electrons to electrode and/or to insoluble Fe (III)-oxides. This was not shown in these papers, which are on cable bacteria motility and on the discovery of the cable bacteria. | Thirdly, we thank the reviewer for pointing out our mistakes when sorting references. We did mean to cite Bjerg et al. 2016 and Pfeffer et al. 2012 in our manuscript. We cited Holmes et al. 2004 to indicate that *D. propionicus* can utilize an electrode as an electron acceptor to oxidize $S^0$, $H_2$, and organic acids like pyruvate, lactate, and propionate. | Thirdly, we will carefully reexamine our referencing to make sure that each citation accurately represents preceding statements. We will also add in new references and revise the introduction to keep our information about the current cable bacteria research up-to-date.
Suggested references have been revised and added.
Line 62 to 63
Line 341
Line 406 to 408
Line 420 to 424
Line 478 to 479
We also added Meysman et al., 2019 and Kjeldsen et al., 2019.
Line 62 to 63
Line 431 to 434
Line 464 to 468 |
| Some statements are highly speculative and not supported by the presented data (l 293) "In summary, when growing on an electrode poised at an oxidative potential, cable bacteria may no longer require long filaments or be able to maintain them due to the nature of the potential gradient" There are no data in the study that can document such statement. | Lastly, the statement pointed out by the reviewer is speculative, though this statement was deduced from observations from our study and another study by Aller et al. 2019. We will make this statement less speculative in revision by only commenting on the observations of Aller et al. | Lastly, we will revise the speculative statement pointed out by the reviewer. We will integrate observations from our study with those presented by Aller et al. 2019 without further speculation.

Line 329 to 335 |

[revised manuscript text omitted]

adapted from © Google Maps 2018

[Figure]

**Fig. 2**

[Figure]

a.

[Figure]

b.

c.

[Figure]

[Figure]

[Figure]

**Fig. 4**

[Figure]

[Figure]

[Figure]

[Figure]

**Fig. 6**

[Figure]

Fig. 7

Fig. 7

[Figure]

[Figure]

---

## Author Response (AR2)

**College of Earth, Ocean, and Atmospheric Sciences**
Oregon State University
104 CEOAS Admin Bldg.
Corvallis, Oregon 97331-5503

**P** 541-737-3504
**F** 541-737-2064
ceoas.oregonstate.edu

12/16/19

Resubmission of the manuscript "Inducing the Attachment of Cable Bacteria on Oxidizing Electrodes (bg-2019-334)"

Dear Editor,

Thank you for the opportunity to let us revise our manuscript: "**Inducing the Attachment of Cable Bacteria on Oxidizing Electrodes (bg-2019-334)**" and consideration for publication. We appreciate the reviewer's comments and suggestions to our manuscript. The revised manuscript is improved after making edits according to reviewer's comments.

Following this letter are our point-to-point response to reviewer's comments including how and where our manuscript has been modified and a no-mark-up version showing the changes made in our manuscript.

Thank you for your time.

Sincerely,

Cheng Li, Ph.D.

| Comments from reviewer #1 | Authors response | Changes in manuscript |
|---|---|---|
| I think that this manuscript has improved significantly in clarity. It is much more targeted and focused than the previous version. I also understand if it is not practical possible to make quantitative estimates of cable bacteria abundance on the electrode. I find that the data presented sufficiently document , that cable bacteria indeed were attached to the electrodes and most likely alive justifying the suggestion that cable bacteria can grow on oxidizing electrodes. I therefore recommend publication. | We thank the reviewer for such positive comments. | No change was proposed. |
| Minors:
l. 62 Larsen and Nielsen 2015 is actually Is actually Larsen, S., L. P. Nielsen, and A. Schramm. 2015. Cable Bacteria Associated with Long Distance Electron Transport in New England Salt Marsh Sediment. Env Microbiol Rep 7: 175-179.
And should be cited as Larsen et. al. 2015 and the reference should be corrected also in the reference list. | we thank the reviewer for pointing out our mistakes when sorting references. We will change this cited article and also do a thoroughly check for our reference. | Cited article has been corrected. And the reference list was revised to ensure that we cite the articles properly.

Line 63

Line 301-395 |
| l. 125: How was the sulfide standard prepared: was it from Na2S crystals: please specify. Further was the sulfide in the calibration series analyzed afterwards by e.g. means of the Cline method? | A sulfide standard of 3 mM was prepared by using $Na_2S.9H_2O$ crystal. We did not perform Cline method to analyze the calibration afterwards. We revised this part of the method to ensure clarity. | Line 125-126 |

---

## Author Response (AR3)

**College of Earth, Ocean, and Atmospheric Sciences**
Oregon State University
104 CEOAS Admin Bldg.
Corvallis, Oregon 97331-5503

**P** 541-737-3504
**F** 541-737-2064
ceoas.oregonstate.edu

1/14/20

Resubmission of the manuscript "Inducing the Attachment of Cable Bacteria on Oxidizing Electrodes (bg-2019-334)"

Dear Editor,

Thank you for accepting our manuscript: "**Inducing the Attachment of Cable Bacteria on Oxidizing Electrodes (bg-2019-334)**". We appreciate the editors' kindness and patient to our manuscript. The revised manuscript is corrected with inserted citation from Geerlings et al, 2019. And we believe it is now ready for publication.

Thank you for your time.

Sincerely,

Cheng Li, Ph.D.